# Burnout among anesthesia personnel in Thailand: A national survey of workload, personality traits, and resilience

Napasorn Sakulteera[1☯], Pongtong Puranitee[2☯], Atipong Pathanasethpong[3☯], Sirirat Rattana-arpa[1☯], Kasana Raksamani[1☯]*

**1** Department of Anesthesiology, Faculty of Medicine Siriraj Hospital, Mahidol University, Bangkok, Thailand, **2** Department of Pediatrics, Faculty of Medicine Ramathibodi Hospital, Mahidol University, Bangkok, Thailand, **3** Department of Anesthesiology, Faculty of Medicine, Khon Kaen University, Khon Kaen, Thailand

☯ These authors contributed equally to this work.
* Kasana.rak@mahidol.ac.th

## Abstract

### Background

Burnout is a critical issue affecting anesthesia personnel, with implications for both provider well-being and patient safety. This study aimed to examine the prevalence of high-risk burnout among anesthesia providers in Thailand and identify associated demographic, occupational, and psychological predictors.

### Methods

A national cross-sectional survey was conducted among anesthesia personnel in Thailand. Burnout was assessed using the Maslach Burnout Inventory–Human Services Survey (MBI-HSS), with high-risk burnout defined by emotional exhaustion ≥27 and depersonalization ≥10. Personality traits were measured using the 20-item Mini International Personality Item Pool (Mini-IPIP), which evaluates five domains: openness, conscientiousness, extraversion, agreeableness, and neuroticism (scored from 1 to 5). Resilience was assessed using the 10-item Connor-Davidson Resilience Scale (CD-RISC-10), with total scores ranging from 0 to 40 and participants categorized into quartiles. Univariable and multivariable logistic regression analyses were used to identify independent predictors.

### Results

Of the 613 responses received, 593 were included in the final analysis. High-risk burnout was identified in 262 participants (44.2%). The high-risk group had significantly higher neuroticism scores (mean 4.11±0.81) and lower scores for conscientiousness (4.21±0.69), extraversion (3.72±0.82), and agreeableness (4.39±0.78) compared to the non-burnout group. No significant difference was found in openness

**Data availability statement:** The data underlying the results presented in this study contain potentially sensitive information from human participants and cannot be shared publicly due to ethical restrictions imposed by the Siriraj Institutional Review Board (SIRB 862/2023). De-identified data are available upon reasonable request from the Siriraj Institutional Review Board, Faculty of Medicine Siriraj Hospital, Mahidol University (contact via: sirb.ec4@gmail.com), for researchers who meet the criteria for access to confidential data.

**Funding:** This research project was supported by the Faculty of Medicine Siriraj Hospital, Mahidol University, under the Grant Number (IO) R016731019. The award was received by K.R. (Kasana Raksamani). Funder: Faculty of Medicine Siriraj Hospital, Mahidol University Grant Number: (IO) R016731019 Funder Website: https://www2.si.mahidol.ac.th/en/ Role of the Funder: The sponsor had no role in the study design, data collection and analysis, decision to publish, or preparation of the manuscript.

**Competing interests:** The authors have declared that no competing interests exist.

scores. Regarding resilience, most high-risk participants (76.3%) fell within the lowest CD-RISC-10 quartile (score range 0–29). Independent predictors of high-risk burnout included higher self-reported burnout severity (adjusted OR 3.84, 95% CI: 2.70–5.47), intention to leave the current role (OR 1.73, 95% CI: 1.21–2.47), higher neuroticism (OR 1.28, 95% CI: 1.13–1.46), and lower resilience (OR 0.70, 95% CI: 0.50–0.99). Other factors such as younger age and frequent night shifts were significant in univariable analysis but not in the final adjusted model.

## Conclusion

Burnout is prevalent among Thai anesthesia personnel and is strongly associated with individual psychological characteristics—particularly higher neuroticism and lower resilience. These findings support the implementation of targeted interventions such as resilience-building programs, psychological screening, and workload management to reduce burnout risk in anesthesia practice.

## Introduction

Burnout is a psychological syndrome that develops in response to chronic interpersonal stressors in the workplace [1]. It is commonly explored through three core dimensions: overwhelming exhaustion, feelings of cynicism and detachment from work, and a sense of ineffectiveness or lack of accomplishment [2]. Burnout can lead to reduced work efficiency, job dissatisfaction, and resignation. It is also associated with serious health consequences, including substance abuse, depressive disorder, and suicidal ideation [2,3]. Both organizational and personal factors contribute to the development of burnout. Organizational factors include job characteristics, professional demands, work environment, and institutional culture. Personal factors encompass age, years of experience, family support, personality traits, stress management abilities, and attitudes toward work [1,2,4,5].

Burnout is more prevalent in healthcare professionals than in other occupational groups [6–9]. Among physicians, anesthesia personnel are particularly vulnerable due to the high-stress nature of their work [8]. The reported prevalence of burnout among anesthesia personnel varies widely—ranging from 12% to 69%—depending on professional roles (residents, anesthesiologists, or nurse anesthetists) and country of origin [10–18]. These differences may stem from variations in regional practices, healthcare systems, workplace cultures, and diagnostic criteria [10–18]. In Thailand, no national data currently exist on burnout prevalence among anesthesia personnel. However, one study reported a burnout rate of 62% among intensivists—a group with work characteristics similar to anesthesia professionals [19].

Individual personality traits have also been identified as significant predictors of burnout. A number of systematic reviews and meta-analyses have shown a consistent association between personality profiles and the risk of burnout [20,21]. The five-factor model (OCEAN)—which includes openness to experience, conscientiousness, extraversion, agreeableness, and neuroticism—is the most widely used

framework in this field [20–22]. A study among Dutch anesthesiologists found that conscientiousness and agreeableness were dominant traits, while a minority exhibiting higher levels of neuroticism—characterized by emotional instability and negative affect—were more prone to burnout [17]. These findings emphasize the moderating role of personality in burnout susceptibility.

Resilience, defined as the ability to adapt to stress and recover from adversity, such as disappointment, failure, or mistakes, is another protective factor against burnout [23]. Resilience is shaped by individual characteristics as well as environmental and cultural influences. Multiple studies have demonstrated its beneficial effects among healthcare workers [23–25]. McCain et al. found that physicians with higher resilience experienced lower burnout and employed more effective coping mechanisms [23]. Similarly, Guo et al. reported a negative association between resilience and burnout in nurses, especially concerning emotional exhaustion and depersonalization [24]. Rushton further observed that nurses in high-stress environments who demonstrated stronger resilience experienced less burnout and had better overall well-being [25].

Given the high-intensity and decision-critical nature of anesthesia work, anesthesia personnel face constant clinical challenges and emergency situations while being expected to maintain alertness and composure. This study focuses on the prevalence of burnout in this professional group. The primary objective is to determine the prevalence of high-risk burnout among anesthesia personnel. The secondary objectives are to examine the associations between burnout, personality traits, and resilience, and to identify key risk factors for high-risk burnout.

## Materials and methods

This nationwide cross-sectional survey was conducted in Thailand between April 10th and June 9th 2024. All anesthesia personnel currently practicing in the field—including anesthesiologists and certified registered nurse anesthetists (CRNAs)—were invited to participate. The online questionnaire was disseminated via social media and through posters mailed to hospitals across the country. Each poster included a QR code linking directly to the survey, which was hosted on Google Forms.

The required sample size was calculated using Elsie Fisher's formula. Based on a previously reported burnout prevalence of 59.2% and a 95% confidence level, a minimum of 368 participants was needed [19]. However, due to the typically low response rates observed in cross-sectional surveys, all eligible anesthesiologists and CRNAs were approached to maximize participation.

The study was approved by the Siriraj Institutional Review Board (approval number: SIRB 862/2023) Participation was entirely voluntary, and responses were collected anonymously. An informed consent statement was provided at the beginning of the questionnaire, explaining the purpose of the study, confidentiality measures, and the right to withdraw at any time. Participants indicated their agreement by proceeding with the survey, as approved by the ethics committee.

The questionnaire consisted of five sections, comprising a total of 67 items. These included: (1) a consent statement, (2) sociodemographic questions, (3) the Maslach Burnout Inventory–Human Services Survey (MBI-HSS), (4) the Mini-International Personality Item Pool (Mini-IPIP), and (5) the 10-item Connor-Davidson Resilience Scale (CD-RISC-10). Validated Thai translations were used for all instruments, except for the Mini-IPIP, which was administered in English due to the lack of a validated Thai version.

### Burnout measurement

Burnout was assessed using the Thai-translated version of the MBI-HSS, which evaluates three core dimensions of burnout: emotional exhaustion (9 items), depersonalization (5 items), and personal accomplishment (8 items). Each item was rated on a 7-point scale ranging from 0 (never) to 6 (every day), and subscale scores were calculated by summing the respective items. To provide a multidimensional characterization of burnout, participants were additionally categorized using the theoretical burnout profile framework described by Maslach and Leiter. This framework classifies individuals

based on whether their scores on emotional exhaustion, depersonalization, and personal accomplishment fall within high or low ranges using established cut-off values. Profiles include Engaged (low EE, low DP, high PA), Ineffective (low EE, low DP, low PA), Overextended (high EE only), Disengaged (high DP only), and Burnout (high EE, high DP, low PA). This classification approach is theory-driven and based on predefined thresholds rather than latent class or cluster-based statistical methods. Additionally, we identified individuals at high risk of burnout, defined as having an emotional exhaustion score ≥27 and/or a depersonalization score ≥10, in order to examine potential risk factors and guide targeted prevention strategies [10].

## Personality traits measurement

Personality traits were assessed using the 20-item Mini-IPIP, which evaluates the five-factor model: openness, conscientiousness, extraversion, agreeableness, and neuroticism (4 items per trait). Items were rated on a 5-point Likert scale from 1 (very inaccurate) to 5 (very accurate). Negatively keyed items were reverse-scored. Trait scores were calculated by summing the items for each domain, with higher scores indicating stronger expressions of the respective trait.

## Resilience measurement

Resilience was measured using the Connor Davidson Resilience Scale (CD-RISC-10), which consists of 10 items reflecting aspects such as flexibility, self-efficacy, emotional regulation, optimism, and attentional control under stress. Responses were rated on a 5-point scale from 0 (not at all true) to 4 (true nearly all the time), with higher total scores indicating greater resilience.

## Statistical analysis

Data were analyzed using IBM SPSS Statistics for Windows, version 29.0 (IBM Corp., Armonk, NY, USA). Descriptive statistics were used to summarize participant characteristics, burnout prevalence, personality traits, and resilience scores. Categorical variables were presented as frequencies and percentages, and continuous variables were reported as means and standard deviations. Missing data were handled using complete case analysis.

Comparisons between participants with and without high-risk burnout were conducted using chi-square tests for categorical variables and independent t-tests for continuous variables. Association between burnout and resilience or personality traits were assessed through group comparisons and logistic regression.

Univariable logistic regression was first used to identify potential risk factors for high-risk burnout. Variables with a p-value <0.10 were entered into a multivariable logistic regression model. Backward elimination using the Wald method was applied, with a removal criterion of p > 0.17, to determine independent predictors. Adjusted odds ratios (ORs) and 95% confidence intervals (CIs) were reported. A two-tailed p-value <0.05 was considered statistically significant.

Participants with incomplete survey responses were excluded from the final analysis through complete case analysis. All statistical tests were two-tailed, and a p-value of less than 0.05 was considered to indicate statistical significance.

# Results

## Participant characteristics

A total of 613 anesthesia personnel completed the survey. After excluding 20 responses with missing data, 593 responses were included in the final analysis. Of these, 262 participants (44%; 95% CI: 39–50%) met the criteria for high-risk burnout, defined by an emotional exhaustion score ≥27 and a depersonalization score ≥10 on the MBI-HSS.

The majority of participants were female (88.0%), and 5.4% identified as LGBTQ+. There were no significant differences in gender or LGBTQ+ identity between those at high risk and those not at risk for burnout (p=0.62). Similarly, marital status and number of children were not significantly associated with burnout risk. Table 1 presents the demographic,

**Table 1. Characteristics of the 593 participants, comparing high-risk and non-high-risk burnout groups.**

| Variable | Category | High-Risk (%) | Non High-Risk (%) | P-value |
|---|---|---|---|---|
| Gender | Male | 5.70 | 7.30 | 0.62 |
| | Female | 88.20 | 87.90 | |
| | LGBTQ+ | 6.10 | 4.80 | |
| Professional Role | CRNA | 78.40 | 68.40 | 0.007 |
| | Anesthesiologist | 21.60 | 31.60 | |
| Age (years) | ≤ 30 | 8.40 | 4.20 | 0.014 |
| | 31–50 | 73.30 | 69.50 | |
| | ≥ 51 | 18.30 | 26.30 | |
| Years of Experience | <10 | 48.10 | 37.60 | 0.015 |
| | 10–30 | 47.30 | 53.90 | |
| | >30 | 4.60 | 8.50 | |
| Marital Status | Single | 43.70 | 40.00 | 0.25 |
| | Married | 48.70 | 54.80 | |
| | Divorced | 5.70 | 3.00 | |
| | Other | 1.90 | 2.10 | |
| No. of Children | 0 | 57.30 | 51.20 | 0.26 |
| | 1 | 19.50 | 18.80 | |
| | 2 | 21.40 | 26.40 | |
| | 3 | 1.90 | 3.60 | |
| Hospital Type | General | 40.00 | 41.00 | 0.007 |
| | University affiliated | 15.80 | 19.80 | |
| | Private | 3.50 | 8.80 | |
| | Regional | 35.80 | 24.60 | |
| | Community | 5.00 | 5.80 | |
| Shift Frequency/Month | None | 15.30 | 25.40 | 0.005 |
| | 1–5 | 19.50 | 23.00 | |
| | 6–10 | 23.40 | 20.50 | |
| | >10 | 41.80 | 31.10 | |
| Working Days/Month | <10 | 0.40 | 1.20 | 0.001 |
| | 10–15 | 1.90 | 4.80 | |
| | 16–20 | 25.00 | 35.60 | |
| | >20 | 72.70 | 58.30 | |
| Income/Month (THB) | ≤ 50,000 | 48.50 | 40.80 | 0.003 |
| | 50,001–100,000 | 37.00 | 33.20 | |
| | >100,000 | 14.50 | 26.00 | |
| Sleep Duration/Day | <4 hrs | 4.20 | 1.20 | 0.008 |
| | 4–6 hrs | 52.70 | 45.60 | |
| | ≥ 7 hrs | 43.10 | 53.20 | |
| Self-Reported Burnout | Lowest–Low | 5.00 | 39.70 | < 0.001 |
| | Mid | 30.50 | 50.00 | |
| | High–Highest | 64.50 | 10.30 | |
| Continue with the Job | Staying | 15.30 | 54.40 | < 0.001 |
| | Unsure | 51.90 | 31.70 | |
| | Quitting | 32.80 | 13.90 | |

professional, and work-related characteristics of the participants included in the final analysis, comparing those classified as high risk for burnout with those not at high risk. Significant differences were observed between the two groups across several variables. High-risk burnout was more prevalent among younger participants (≤30 years), those with fewer than 10 years of work experience, and certified registered nurse anesthetists (CRNAs). Personnel working in regional hospitals, performing more frequent night shifts, or working over 20 days per month also demonstrated higher burnout rates. Additionally, lower income levels, shorter sleep duration, and self-reported high levels of burnout were significantly associated with high-risk status. Career intentions differed notably, with those planning to leave their current roles more likely to be in the high-risk group.

Burnout profiles were assigned using the predefined Maslach theoretical framework and are presented descriptively to illustrate patterns of burnout dimensions in the sample. The largest proportion of participants (42.8%) were classified as ***Engaged***, indicating low levels of exhaustion and cynicism along with high personal efficacy, reflecting a healthy state of work engagement. ***Ineffective*** individuals accounted for 25.3% of the sample, characterized by preserved emotional stability but reduced feelings of accomplishment. ***Mixed*** profiles, which did not align clearly with a single burnout pattern, made up 13.5% of respondents. Meanwhile, 10.1% fell into the ***Overextended*** category, showing high exhaustion likely related to workload demands. The most severe form of burnout, the ***Burnout*** profile—defined by high exhaustion, high cynicism, and low efficacy—was observed in 7.3% of participants. Lastly, a small proportion (1.0%) were classified as ***Disengaged***, indicating emotional withdrawal or lack of motivation without accompanying exhaustion (Table 2).

## Factors associated with high-risk burnout

Several demographic and work-related variables were significantly associated with high-risk burnout. Younger participants (≤30 years) were more likely to experience burnout compared to their older counterparts (8.4% vs. 4.2%, p = 0.014). Participants with fewer than 10 years of work experience had a higher prevalence of burnout than those with longer tenure (48.1% vs. 37.6%, p = 0.015).

Professional role was also associated with burnout risk: certified registered nurse anesthetists (CRNAs) had a significantly higher prevalence of high-risk burnout than anesthesiologists (78.4% vs. 68.4%, p = 0.007). In terms of workplace setting, personnel employed in provincial hospitals reported higher burnout rates compared to those in university-affiliated or private hospitals (p = 0.007).

Workload factors further influenced burnout risk. Those working more than 10 night shifts per month were more likely to experience high-risk burnout (p = 0.005). Income level showed an inverse relationship, with participants earning ≤50,000 THB per month experiencing significantly higher burnout rates (p = 0.003). Sleep deprivation was also a contributing factor: individuals sleeping fewer than four hours per day were more likely to be in the high-risk group (p = 0.008).

Self-perceived burnout aligned with the objective measures: 64.5% of those who rated their burnout as "high to highest" were in the high-risk group, compared to only 10.3% in the non-burnout group (p < 0.001). Additionally, only 15.3% of

**Table 2. Distribution of burnout profiles based on MBI classification.**

| Profile | Description | N | (%) |
|---|---|---|---|
| Engaged | Low exhaustion, low cynicism, high efficacy – reflects healthy work engagement | 254 | (42.8%) |
| Ineffective | Low exhaustion and cynicism, but low efficacy – signals reduced confidence or satisfaction | 150 | (25.3%) |
| Mixed | Does not fit neatly into any single profile – presents a blend of symptoms | 80 | (13.5%) |
| Overextended | High exhaustion only – often due to excessive workload | 60 | (10.1%) |
| Burnout | High exhaustion, high cynicism, and low efficacy – full burnout syndrome | 43 | (7.3%) |
| Disengaged | High cynicism only – detachment and lack of motivation | 6 | (1.0%) |

high-risk participants intended to remain in their current roles long-term, while over half (54.4%) of the non-burnout group planned to stay (p < 0.001).

## Personality traits and burnout (Mini-IPIP)

Personality was assessed using the Mini-International Personality Item Pool (Mini-IPIP), evaluating the five-factor model: openness, conscientiousness, extraversion, agreeableness, and neuroticism. Participants showed relatively high mean scores in conscientiousness (4.29 ± 0.70) and agreeableness (4.51 ± 0.87), moderate scores in openness (4.45 ± 0.80) and extraversion (3.83 ± 0.80), and lower scores in neuroticism (3.81 ± 0.81). This profile reflects the organized, cooperative, and emotionally stable characteristics often found in high-responsibility healthcare roles.

Significant differences in personality traits were observed between high-risk and non-high-risk groups (Table 3). The high-risk group had significantly higher neuroticism scores (4.11 ± 0.81 vs. 3.58 ± 0.73; p < 0.001) and lower scores in conscientiousness (4.21 ± 0.69 vs. 4.36 ± 0.71; p = 0.012), extraversion (3.72 ± 0.82 vs. 3.91 ± 0.77; p = 0.005), and agreeableness (4.39 ± 0.78 vs. 4.61 ± 0.92; p = 0.002). No significant difference was found in openness scores (p = 0.098). These findings suggest that higher neuroticism and lower levels of positive personality traits may contribute to burnout vulnerability among anesthesia personnel.

## Resilience and burnout

Resilience was measured using the 10-item Connor-Davidson Resilience Scale (CD-RISC-10), and participants were stratified into quartiles. A strong inverse relationship between resilience and burnout was observed (Table 4). The lowest resilience quartile (scores 0–29) comprised 62.9% of the sample and accounted for 76.3% of those at high risk for burnout (p < 0.001). In contrast, only 5.7% of those in the highest resilience quartile (scores 37–40) were classified as high

**Table 3. Comparison of MINI-IPIP personality trait scores between high-risk and non-high-risk burnout groups (n = 593).**

| Personality Trait | Mean ± SD (Total) | High-Risk Burnout | Non-High-Risk Burnout | p-value |
|---|---|---|---|---|
| Openness | 4.45 ± 0.80 | 4.39 ± 0.79 | 4.50 ± 0.81 | 0.098 |
| Conscientiousness | 4.29 ± 0.70 | 4.21 ± 0.69 | 4.36 ± 0.71 | 0.012 |
| Extraversion | 3.83 ± 0.80 | 3.72 ± 0.82 | 3.91 ± 0.77 | 0.005 |
| Agreeableness | 4.51 ± 0.87 | 4.39 ± 0.78 | 4.61 ± 0.92 | 0.002 |
| Neuroticism | 3.81 ± 0.81 | 4.11 ± 0.81 | 3.58 ± 0.73 | <0.001 |

Note. Higher scores reflect greater expression of each trait. Trait definitions based on the Five-Factor Model:- Openness: Imagination, wide interests- Conscientiousness: Organization, self-discipline- Extraversion: Sociability, energy, positive affect- Agreeableness: Forgiveness, modesty, straightforwardness- Neuroticism: Anxiety, irritability, emotional instability.

**Table 4. Distribution of Resilience Levels (CD-RISC-10 Quartiles) by Burnout Risk Status (n = 593).**

| CD-RISC-10 Resilience Level | Total n (%) | High-Risk Burnout n (%) | Non-High-Risk Burnout n (%) | p-value |
|---|---|---|---|---|
| Lowest Quartile (0–29) | 371 (62.9%) | 200 (76.3%) | 171 (52.1%) | <0.001 |
| Second Quartile (30–32) | 99 (16.8%) | 30 (11.5%) | 69 (21.0%) | |
| Third Quartile (33–36) | 74 (12.5%) | 17 (6.5%) | 57 (17.4%) | |
| Top Quartile (37–40) | 46 (7.8%) | 15 (5.7%) | 31 (9.5%) | |

Note. Resilience was assessed using the 10-item Connor-Davidson Resilience Scale (CD-RISC-10), with participants categorized into quartiles based on total scores. The lowest quartile (0–29) represents the lowest level of resilience. A significant inverse association was observed between resilience and high-risk burnout (p < 0.001).

risk. The prevalence of burnout progressively declined with increasing resilience scores, highlighting the protective role of resilience in mitigating burnout.

## Multivariable analysis

Multivariable logistic regression analyses were conducted using the predefined high-risk burnout definition (emotional exhaustion ≥27 and/or depersonalization ≥10) as the binary outcome variable. After adjusting for confounding variables, higher self-reported burnout severity (adjusted OR 3.84, 95% CI: 2.70–5.47, p < 0.001), intention to leave one's current job (adjusted OR 1.73, 95% CI: 1.21–2.47, p = 0.003), and higher neuroticism scores (adjusted OR 1.28, 95% CI: 1.13–1.46, p < 0.001) were significantly associated with increased burnout risk. Conversely, greater resilience was associated with lower risk (adjusted OR 0.70, 95% CI: 0.50–0.99, p = 0.044) (Table 5). Other variables, including younger age, hospital type, number of shifts and working days per month, showed significant associations in univariable analyses but were not retained as independent predictors in the final model. The final model demonstrated good discriminatory performance, with an area under the receiver operating characteristic curve (AUC) of 0.868 (95% CI: 0.84–0.89).

## Discussion

This study highlights a substantial prevalence of high-risk burnout among anesthesia personnel in Thailand, with 44% of participants meeting the criteria based on the Maslach Burnout Inventory–Human Services Survey (MBI-HSS). Multivariable logistic regression identified several independent predictors of high-risk burnout, including higher self-reported burnout severity, the intention to leave anesthesia-related work, higher neuroticism scores, and lower resilience. While factors such as younger age, increased shift frequency, and number of working days were significant in univariable analysis, they did not remain significant in the final multivariable model. These findings underscore the complex interplay between occupational stress, individual psychological traits, and career-related attitudes in the development of burnout.

The observed prevalence of high-risk burnout (44.2%) is consistent with international reports. For example, a 2020 U.S. study found that 59.2% of anesthesiologists were at risk of burnout, with key contributors including long working hours, staffing shortages, and poor work-life balance [8,10]. Our study identified four independent predictors of high-risk burnout: higher self-reported burnout severity, the intention to leave one's current job, higher neuroticism, and lower resilience. Among these, self-reported burnout severity emerged as the strongest predictor in the multivariable model. This finding is consistent with prior research by Eckleberry-Hunt et al., who demonstrated that self-reported burnout among physicians significantly correlated with diminished wellness and perceived quality of patient care [26]. Their findings reinforce the validity and clinical relevance of subjective burnout assessments, highlighting their predictive value beyond traditional occupational metrics. Additionally, the intention to leave one's current role was associated with a 1.7-fold increased likelihood of high-risk burnout, underscoring the psychological toll and dissatisfaction experienced by at-risk personnel [27].

**Table 5. Multivariable logistic regression analysis of factors associated with high-risk burnout.**

| Variable | Crude OR (95% CI) | Crude p-value | Adjusted OR (95% CI) | Adjusted p-value |
|---|---|---|---|---|
| Age group (younger vs older) | 1.78 (1.15–2.76) | 0.01 | 0.62 (0.37–1.04) | 0.067 |
| Hospital type | 1.32 (1.02–1.72) | 0.03 | 1.12 (0.96–1.31) | 0.154 |
| Number of shifts per month | 1.58 (1.20–2.09) | 0.002 | 1.23 (0.98–1.55) | 0.079 |
| Number of working days per month | 1.66 (1.12–2.45) | 0.015 | 1.47 (0.95–2.26) | 0.081 |
| Self-reported burnout severity | 5.23 (3.15–8.69) | <0.001 | 3.84 (2.70–5.47) | <0.001 |
| Intention to stay in current job | 2.61 (1.91–3.56) | <0.001 | 1.73 (1.21–2.47) | 0.003 |
| Neuroticism score | 1.45 (1.21–1.74) | <0.001 | 1.28 (1.13–1.46) | <0.001 |
| Resilience score | 0.68 (0.52–0.89) | 0.004 | 0.70 (0.50–0.99) | 0.044 |

Consistent with previous research, our study found that specific personality traits were significantly associated with burnout risk. Individuals with higher levels of neuroticism—and lower levels of conscientiousness, extraversion, and agreeableness—were more likely to experience high-risk burnout. These results align with a study of Dutch anesthesiologists, which identified neuroticism as the strongest predictor of burnout and noted protective effects from conscientiousness and extraversion [17]. A systematic review of studies using the Five-Factor personality model further reinforced these associations, identifying neuroticism as the most consistent predictor of burnout across occupational groups, while conscientiousness and extraversion were consistently associated with lower burnout risk [20].

In our multivariable analysis, neuroticism remained an independent predictor of burnout, suggesting that psychological vulnerability—particularly difficulty with emotional regulation and stress tolerance—plays a key role in burnout development [20]. Interestingly, openness to experience was not significantly associated with burnout in our sample, which contrasts with the Dutch study's findings linking openness with lower burnout [17]. This divergence may reflect cultural or contextual differences in how traits like openness influence coping styles and adaptability in the workplace.

In the present study, higher resilience among anesthesia personnel was associated with significantly lower burnout levels. This aligns with prior findings in health professions, where individuals with greater resilience have demonstrated markedly lower emotional exhaustion and depersonalization scores [28,29]. A U.S. survey showed that higher resilience scores were associated with lower rates of burnout among physicians and the general workforce, although even highly resilient individuals were not immune to workplace stressors [28]. Similarly, in anesthesiology, resilience has been described as a dynamic quality that can buffer the effects of stress and emotional exhaustion, suggesting its value as a target for wellness interventions [30]. These findings underscore the importance of resilience-building interventions, such as mindfulness training, coping skills workshops, and institutional wellness programs, in mitigating burnout risks [4,28,29]. Together, these findings reinforce a multifactorial model of burnout in anesthesia personnel, shaped by both occupational demands and individual psychological profiles.

Notably, Certified Registered Nurse Anesthetists (CRNAs) exhibited a higher prevalence of high-risk burnout compared to anesthesiologists in our study. This finding is consistent with previous literature demonstrating that CRNAs are particularly vulnerable to burnout due to a combination of high workload, limited autonomy, and the emotional demands of their role [11,27]. An integrative review by Del Grosso and Boyd emphasized that CRNAs face intense job-related stressors, including long hours, complex clinical tasks, and evolving healthcare demands, which contribute to burnout at rates comparable to or exceeding those of physicians in similar settings [31]. Furthermore, studies have shown that CRNAs often report higher emotional exhaustion and lower job control compared to their physician counterparts, suggesting systemic and structural contributors to their burnout risk [31]. These observations align with our findings and underscore the need for targeted interventions to support the well-being of nurse anesthetists within multidisciplinary anesthesia teams.

## Implications and recommendations

The findings of this study highlight several actionable strategies to mitigate burnout among anesthesia personnel. First, screening for personality traits—particularly high neuroticism and low conscientiousness—early in training or professional development may help identify individuals at elevated risk, enabling targeted psychological support or mentoring. Second, implementing structured resilience-building programs, such as mindfulness-based stress reduction or coping skills workshops, may offer meaningful protection against burnout, as supported by both our results and prior research. Lastly, efforts to manage workload—especially by reducing excessive shift frequency—could alleviate occupational stress. Special attention should be given to high-risk groups, including CRNAs and younger personnel, who may benefit from tailored interventions and organizational support. Together, these approaches underscore the need for both individual-focused and system-level strategies to foster well-being and prevent burnout in the anesthesia workforce.

**Strengths and limitations**

This study offers several strengths. It is one of the few investigations to explore burnout among anesthesia personnel in Thailand using a large, diverse, and well-defined national sample. The use of the Maslach Burnout Inventory–Human Services Survey (MBI-HSS), a widely validated and internationally recognized instrument, enhances the credibility and comparability of the findings. Furthermore, the inclusion of psychological factors such as personality traits (via the Mini-IPIP) and resilience (via CD-RISC-10), alongside detailed demographic and occupational variables, allows for a comprehensive, multifactorial analysis of burnout risk. The use of multivariable logistic regression with backward elimination model refinement strengthens the identification of independent predictors.

Nonetheless, several limitations should be acknowledged. Although validated instruments were used, all measures were self-administered and therefore subject to social desirability bias and potential response distortion. In addition, cultural and contextual factors within the Thai healthcare system may influence how burnout, personality traits, and resilience are perceived and reported, which may affect the interpretation and generalizability of the findings.

The cross-sectional design limits causal inference, as temporal relationships between predictors and burnout cannot be established. While the study included a large nationwide sample of anesthesia personnel, the findings may not be directly generalizable to healthcare systems with different organizational structures, workload patterns, or cultural norms.

Furthermore, certain potentially influential variables—such as leadership style, institutional wellness initiatives, and interprofessional dynamics—were not captured and may contribute to burnout risk. Although multivariable models adjusted for several confounders, residual confounding cannot be entirely excluded.

Future research using longitudinal designs or mixed-methods approaches may help clarify causal pathways and evaluate the effectiveness of targeted prevention strategies over time.

## Conclusion

Burnout remains a pressing concern among anesthesia personnel in Thailand, driven by both systemic and individual factors. Our results echo findings from other countries, reinforcing the role of neuroticism, low conscientiousness, and low resilience in increasing vulnerability to burnout. Addressing these factors through institutional policies, tailored support strategies, and psychological skill development may contribute to more sustainable and supportive work environments for anesthesia professionals.

## Author contributions

**Conceptualization:** Napasorn Sakulteera, Pongtong Puranitee, Atipong Pathanasethpong, Sirirat Rattana-arpa, Kasana Raksamani.

**Data curation:** Napasorn Sakulteera, Kasana Raksamani.

**Formal analysis:** Napasorn Sakulteera, Kasana Raksamani.

**Funding acquisition:** Napasorn Sakulteera, Kasana Raksamani.

**Investigation:** Napasorn Sakulteera, Pongtong Puranitee, Kasana Raksamani.

**Methodology:** Napasorn Sakulteera, Pongtong Puranitee, Atipong Pathanasethpong, Sirirat Rattana-arpa, Kasana Raksamani.

**Project administration:** Napasorn Sakulteera, Pongtong Puranitee, Kasana Raksamani.

**Supervision:** Pongtong Puranitee, Atipong Pathanasethpong, Sirirat Rattana-arpa, Kasana Raksamani.

**Validation:** Pongtong Puranitee, Atipong Pathanasethpong, Sirirat Rattana-arpa, Kasana Raksamani.

**Visualization:** Pongtong Puranitee, Atipong Pathanasethpong, Sirirat Rattana-arpa, Kasana Raksamani.

**Writing – original draft:** Napasorn Sakulteera, Pongtong Puranitee, Atipong Pathanasethpong, Sirirat Rattana-arpa, Kasana Raksamani.

**Writing – review & editing:** Napasorn Sakulteera, Pongtong Puranitee, Atipong Pathanasethpong, Sirirat Rattana-arpa, Kasana Raksamani.

## Acknowledgments

The authors would like to express their sincere gratitude to Dr. Orawan Supapueng for her invaluable assistance with statistical analysis and interpretation. We also thank Ms. Chusana Rungjindamai for her support in coordinating the research process and ensuring smooth study conduct. Lastly, we are deeply grateful to all anesthesia personnel who participated in the survey and contributed their time and insights to this study.

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
