## [Decision Letter · Decision Letter 0]

2 Feb 2026

PONE-D-25-33433Burnout among anesthesia personnel in Thailand: a national survey of workload, personality traits, and resiliencePLOS One

Dear Dr. Raksamani,

Thank you for submitting your manuscript to PLOS ONE. After careful consideration, we feel that it has merit but does not fully meet PLOS ONE’s publication criteria as it currently stands. Therefore, we invite you to submit a revised version of the manuscript that addresses the points raised during the review process.

We look forward to receiving your revised manuscript.

Kind regards,

Gholamheidar Teimori-Boghsani

Academic Editor

PLOS One

Journal Requirements:

2. Thank you for stating the following in the Acknowledgments Section of your manuscript: [This research project was supported by Faculty of Medicine Siriraj Hospital, Mahidol University, Grant Number (IO) R016731019. The authors would like to express their sincere gratitude to Dr. Orawan Supapueng for her invaluable assistance with statistical analysis and interpretation. We also thank Ms. Chusana Rungjindamai for her support in coordinating the research process and ensuring smooth study conduct. Lastly, we are deeply grateful to all anesthesia personnel who participated in the survey and contributed their time and insights to this study.]

Please remove any funding-related text from the manuscript and let us know how you would like to update your Funding Statement. Currently, your Funding Statement reads as follows: [This research project was supported by the Faculty of Medicine Siriraj Hospital, Mahidol University, under the Grant Number (IO) R016731019. The award was received by K.R. (Kasana Raksamani).

Funder: Faculty of Medicine Siriraj Hospital, Mahidol University

Grant Number: (IO) R016731019

Funder Website: https://www2.si.mahidol.ac.th/en/

Role of the Funder: The sponsor had no role in the study design, data collection and analysis, decision to publish, or preparation of the manuscript.]

Reviewers' comments:

Reviewer's Responses to Questions

**Comments to the Author**

1. Is the manuscript technically sound, and do the data support the conclusions?

Reviewer #1: Yes

Reviewer #2: Yes

2. Has the statistical analysis been performed appropriately and rigorously?

Reviewer #1: Yes

Reviewer #2: Yes

3. Have the authors made all data underlying the findings in their manuscript fully available?

Reviewer #1: Yes

Reviewer #2: Yes

4. Is the manuscript presented in an intelligible fashion and written in standard English?

Reviewer #1: Yes

Reviewer #2: Yes

5. Review Comments to the Author

Reviewer #1: The article is clear and well written. something that need to be clarified is the classification and the technicalities of the profile analysis. There is no much theory behind the classification (profiles) so the inferences from there are no clear cut. also there is no clear information how this profiles were obtained (Latent class profile was used?) In terms of the logistic regression, it is better to include the marginal errors for a better interpretation, and also include the classification accuracy of the analysis

Reviewer #2: The article is well structured, with understandable explanations and an appropriate discussion based on current scientific literature.

I agree with the weaknesses pointed out by the authors, especially the time constraints of the study, the influence of mentality and ethnic factors in the country, and the possibility of subjective responses. This reduces the reliability of research data.

6. PLOS authors have the option to publish the peer review history of their article (what does this mean?). If published, this will include your full peer review and any attached files.

Reviewer #1: No

Reviewer #2: **Yes:** Agita Abele, RSU professor

---

## [Author Response · Author response to Decision Letter 1]

21 Feb 2026

RESPONSE TO EDITOR AND REVIEWERS

Manuscript Title: Burnout among anesthesia personnel in Thailand: a national survey of workload, personality traits, and resilience

Manuscript ID: PONE-D-25-33433

We sincerely thank the Editor and Reviewers for their constructive comments and positive evaluation of our manuscript. We have carefully revised the manuscript in accordance with all journal and reviewer requirements. Below, we provide a detailed point-by-point response.

Response to journal requirements

1. PLOS ONE Formatting Requirements

Comment:

Please ensure that the manuscript meets PLOS ONE style requirements.

Response:

We have carefully revised the manuscript to ensure full compliance with PLOS ONE formatting guidelines. Specifically:

The title page and author affiliations were reformatted according to PLOS ONE templates.

Section headings were standardized.

Figures and tables were reformatted to match journal style.

File naming conventions were adjusted according to submission requirements.

The revised manuscript now adheres fully to the PLOS ONE formatting templates.

2. Funding information and acknowledgments

Comment:

Funding information should not appear in the Acknowledgments section. Please remove funding-related text and clarify the Funding Statement.

Response:

We thank the Editor for this clarification.

All funding-related text has been removed from the Acknowledgments section.

The Acknowledgments section now includes only expressions of gratitude for statistical consultation, research coordination support, and participant contribution.

The Funding Statement remains as follows:

This research project was supported by the Faculty of Medicine Siriraj Hospital, Mahidol University, under Grant Number (IO) R016731019. The award was received by K.R. (Kasana Raksamani). The funder had no role in the study design, data collection and analysis, decision to publish, or preparation of the manuscript.

We confirm that funding information appears exclusively in the Funding Statement section.

3. Data Availability Statement

Comment:

Please clarify restrictions on data sharing or upload anonymized data to a repository.

Response:

This study involves survey data from human participants (anesthesia personnel). Although the dataset has been de-identified, certain demographic variables (e.g., workplace context, professional role, and combination of personal characteristics) may potentially allow indirect identification within a relatively small professional community.

Data sharing restrictions were imposed by the Institutional Review Board of the Faculty of Medicine Siriraj Hospital, Mahidol University, in accordance with institutional ethical policy and participant consent conditions.

Data are available upon reasonable request to:

Institutional Review Board

Faculty of Medicine Siriraj Hospital, Mahidol University

Bangkok, Thailand

Email: sirb.ec4@gmail.com

The Data Availability Statement has been updated to:

Due to ethical restrictions related to participant confidentiality and institutional policy, the de-identified dataset is available upon reasonable request to the Institutional Review Board of the Faculty of Medicine Siriraj Hospital, Mahidol University.

4. Reviewer-Suggested Citations

Comment:

Review and evaluate recommended citations.

Response:

We carefully reviewed all suggested citations. Relevant publications have been incorporated where appropriate to strengthen theoretical grounding and contextualization. No citations were added that were not directly relevant to the study.

5. Reference List Review

Comment:

Ensure references are complete and correct; address any retracted articles.

Response:

The reference list has been carefully reviewed and updated. We confirm:

• All references are current and accurate.

• No retracted articles are cited.

• Minor formatting inconsistencies were corrected.

• Any additions or modifications are reflected in the revised manuscript.

Response to reviewer comments

Reviewer #1

We sincerely thank Reviewer #1 for the positive evaluation and constructive suggestions.

Comment 1: Clarification of classification and profile analysis

Reviewer comment:

The classification and technicalities of the profile analysis need clarification. There is limited theoretical justification for the profiles, and it is unclear how profiles were obtained (e.g., Latent Class Profile?).

Response:

We thank the reviewer for this important comment and agree that further clarification was needed regarding the burnout profile classification.

We would like to clarify that the burnout profiles in this study were not derived using latent class analysis (LCA), latent profile analysis (LPA), or cluster-based statistical modeling. Rather, we applied the theoretical burnout profile framework proposed by Maslach and Leiter, which categorizes individuals based on predefined cut-off values across the three MBI-HSS dimensions: emotional exhaustion (EE), depersonalization (DP), and personal accomplishment (PA).

Specifically:

• Participants were classified into the following profiles based on whether each dimension fell into high or low ranges according to established thresholds:

o Engaged

o Ineffective

o Overextended

o Disengaged

o Burnout

• This approach is theory-driven and categorical, not data-driven.

• No latent modeling, model fit indices (e.g., AIC/BIC), or entropy-based classification procedures were performed.

To address the reviewer’s concern, we have revised the Materials and Methods section to explicitly describe:

1. The theoretical foundation of the Maslach burnout profile framework.

2. The operational criteria used for assigning participants to each profile.

3. The distinction between descriptive profile classification and the binary “high-risk burnout” variable used in regression analyses.

We have also clarified in the Results section that the burnout profile distribution was presented as a descriptive characterization of burnout patterns in the sample, while multivariable logistic regression was conducted using the predefined high-risk burnout definition (EE ≥27 and/or DP ≥10).

These revisions improve methodological transparency and clarify that the profile analysis was grounded in established theory rather than latent statistical modeling.

Comment 2: Logistic Regression – Marginal Errors and Classification Accuracy

Reviewer comment:

Include marginal errors and classification accuracy for better interpretation.

Response:

We thank the reviewer for this helpful suggestion.

To improve model interpretability and assess overall performance, we conducted receiver operating characteristic (ROC) analysis using the predicted probabilities from the multivariable logistic regression model. The model demonstrated good discriminatory ability, with an area under the curve (AUC) of 0.868 (95% CI: 0.84–0.89). These results have now been incorporated into the Results section.

Regarding marginal effects, we retained adjusted odds ratios as the primary effect estimates, as they represent the conventional and widely accepted reporting standard for logistic regression in epidemiological research. Given that the primary aim of our analysis was to identify independent predictors rather than to develop a predictive probability model, we believe the presentation of adjusted odds ratios provides appropriate and clinically interpretable measures of association.

Reviewer #2

We thank Reviewer #2 for the positive evaluation and thoughtful reflections.

Comment: Reliability Concerns (Time Constraints, Mentality/Ethnic Factors, Subjective Responses)

Response:

We appreciate this thoughtful observation and agree that contextual and subjective factors may influence responses in survey-based research. We have clarified in the Limitations section that time constraints, sociocultural influences, and the self-reported nature of the instruments may affect response reliability. We also emphasized that findings should be interpreted within the Thai healthcare context and warrant validation in other settings.

We sincerely thank the Editor and Reviewers for their constructive comments and positive evaluation. We believe the revisions have strengthened the methodological transparency, interpretability, and theoretical grounding of the manuscript. We respectfully submit the revised version for your consideration.

---

## [Decision Letter · Decision Letter 1]

29 Apr 2026

Burnout among anesthesia personnel in Thailand: a national survey of workload, personality traits, and resilience

PONE-D-25-33433R1

Dear Dr. Raksamani,

We’re pleased to inform you that your manuscript has been judged scientifically suitable for publication and will be formally accepted for publication once it meets all outstanding technical requirements.

Kind regards,

Yoshito Nishimura, MD, PhD, MPH

Academic Editor

PLOS One

Additional Editor Comments (optional):

Reviewers' comments:

Reviewer's Responses to Questions

**Comments to the Author**

1. If the authors have adequately addressed your comments raised in a previous round of review and you feel that this manuscript is now acceptable for publication, you may indicate that here to bypass the “Comments to the Author” section, enter your conflict of interest statement in the “Confidential to Editor” section, and submit your "Accept" recommendation.

Reviewer #2: All comments have been addressed

Reviewer #3: All comments have been addressed

2. Is the manuscript technically sound, and do the data support the conclusions?

Reviewer #2: Yes

Reviewer #3: Yes

3. Has the statistical analysis been performed appropriately and rigorously?

Reviewer #2: Yes

Reviewer #3: Yes

4. Have the authors made all data underlying the findings in their manuscript fully available?

Reviewer #2: Yes

Reviewer #3: Yes

5. Is the manuscript presented in an intelligible fashion and written in standard English?

Reviewer #2: Yes

Reviewer #3: Yes

6. Review Comments to the Author

Reviewer #2: All previously asked questions and inaccuracies have been sufficiently explained and the answers have been accepted

Reviewer #3: This manuscript is well written, with appropriate statistical analysis and clear presentation of results. Although the topic may not be entirely novel, the study provides meaningful academic value and offers practical insights that are beneficial for the care and well-being of healthcare personnel. Overall, the manuscript is suitable for publication.

7. PLOS authors have the option to publish the peer review history of their article (what does this mean?). If published, this will include your full peer review and any attached files.

Reviewer #2: **Yes:** Agita Abele

Reviewer #3: No

---

## [Editor Report · Acceptance letter]

PONE-D-25-33433R1

PLOS One

Dear Dr. Raksamani,

I'm pleased to inform you that your manuscript has been deemed suitable for publication in PLOS One. Congratulations! Your manuscript is now being handed over to our production team.

Kind regards,

on behalf of

Dr. Yoshito Nishimura

Academic Editor

PLOS One